# Distinct clonal evolution of B-cells in HIV controllers with neutralizing antibody breadth

**Deniz Cizmeci[1], Giuseppe Lofano[1], Evan Rossignol[1], Anne-Sophie Dugast[1], Dongkyoon Kim[2], Guy Cavet[2], Ngan Nguyen[2], Yann Chong Tan[2†], Michael S Seaman[3], Galit Alter[1], Boris Julg[1]***

[1]Ragon Institute of MGH, MIT and Harvard, Cambridge, United States; [2]Atreca Inc, Redwood City, United States; [3]Center for Virology and Vaccine Research, Beth Israel Deaconess Medical Center, Boston, United States

**Abstract** A minor subset of individuals infected with HIV-1 develop antibody neutralization breadth during the natural course of the infection, often linked to chronic, high-level viremia. Despite significant efforts, vaccination strategies have been unable to induce similar neutralization breadth and the mechanisms underlying neutralizing antibody induction remain largely elusive. Broadly neutralizing antibody responses can also be found in individuals who control HIV to low and even undetectable plasma levels in the absence of antiretroviral therapy, suggesting that high antigen exposure is not a strict requirement for neutralization breadth. We therefore performed an analysis of paired heavy and light chain B-cell receptor (BCR) repertoires in 12,591 HIV-1 envelope-specific single memory B-cells to determine alterations in the BCR immunoglobulin gene repertoire and B-cell clonal expansions that associate with neutralizing antibody breadth in 22 HIV controllers. We found that the frequency of genomic mutations in IGHV and IGLV was directly correlated with serum neutralization breadth. The repertoire of the most mutated antibodies was dominated by a small number of large clones with evolutionary signatures suggesting that these clones had reached peak affinity maturation. These data demonstrate that even in the setting of low plasma HIV antigenemia, similar to what a vaccine can potentially achieve, BCR selection for extended somatic hypermutation and clonal evolution can occur in some individuals suggesting that host-specific factors might be involved that could be targeted with future vaccine strategies.

**\*For correspondence:**
bjulg@mgh.harvard.edu

**Present address:** †Esco Ventures X, Singapore, Singapore

## Introduction

The induction of cross-reactive broadly neutralizing antibodies (bNAbs) that can effectively neutralize a majority of circulating HIV-1 strains is a major goal of current HIV vaccine development, but no vaccine candidate to date has achieved this goal (*Burton and Hangartner, 2016*). During natural HIV-1 infection, development of neutralizing antibody responses is seen in about 10–30% of HIV-1-infected individuals but often requires years of infection (*Pilgrim et al., 1997*). Specifically, high viremia (*Mikell et al., 2011*; *CAPRISA002 Study Team et al., 2011*) with rapid viral diversification (*NISC Comparative Sequencing Program et al., 2014*; *NISC Comparative Sequencing Program et al., 2013*) along with infection-associated immune activation (*Sather et al., 2009*) have been associated with the evolution of antibody neutralization breadth. One specific feature that characterizes many bNAbs is the high mutation frequency in their variable domains, indicative of many cycles of somatic hypermutation (SHM) in immunoglobulin (Ig) variable-diversity-joining [V(D)J] gene segments, ultimately resulting in superior B-cell receptor-antigen affinity maturation (*Victora and Nussenzweig, 2012*). The requirement for repeated

cycling through the germinal centers (GCs) to accumulate affinity and specificity enhancing mutations is a lengthy process and is reflected by the time required for bNAbs to be elicited in the presence of a constantly evolving virus. However, bNAbs with high potency and exquisite breadth against cross-clade viral isolates have also been isolated from HIV-infected individuals who spontaneously control HIV to low plasma levels in the absence of antiretroviral therapy ('HIV controllers') (*Walker et al., 2009*; *Simek et al., 2009*; *González et al., 2018*; *Schoofs et al., 2019*; *Freund et al., 2017*), thus suggesting that neutralizing antibody breadth may be achievable in the absence of high levels of viremia and rapidly evolving viral diversity, a scenario that more likely can be accomplished with vaccination strategies. In these HIV controller individuals, evolution of neutralization breadth was linked to low but persistent HIV viral antigenemia in the setting of a unique inflammatory profile, while antigenic diversity was not a significant contributor (*Dugast et al., 2017*). Thus, neutralizing antibody breadth may be achievable in the absence of high levels of viremia and rapidly evolving viral diversity, therefore suggesting a scenario that more likely can be accomplished with vaccination strategies. In this study, we selected 22 controllers who exhibited different levels of neutralization breadth (ranging from 0/11 to 11/11 of tier 2/3 viruses neutralized) (*Freund et al., 2017*; *Dugast et al., 2017*) to elucidate features of the HIV-specific antibody repertoire that are associated with neutralization breadth. We performed a natively paired heavy and light chain B-cell receptor repertoire analysis of HIV-1 envelope (Env)-specific memory B-cells (MBCs) using high-throughput single-cell sequencing to determine alterations in the BCR Ig gene repertoire and B-cell clonal expansions that track with neutralization breadth.

## Results

### Study cohort characteristics

Twenty-two HIV-infected individuals, 4 female and 18 male, were included in this study. 11 participants were African American and 11 were of European descent. All individuals were US-based and therefore likely infected with clade B viruses. Plasma samples were tested for neutralizing activity using a standard reference panel of tier 2 and 3 clade B Env pseudoviruses (*Li et al., 2005*). Six individuals neutralized >90% of viruses at 50% inhibitory dose (ID50) titers above background, and were classified as top-neutralizer (TN), while seven individuals neutralized <10% of the panel viruses and were classified as non-neutralizer (NN). The remaining individuals were considered intermediate neutralizer, exhibiting neutralization of 36–82% of the virus panel (*Supplementary file 1*). There was no statistically significant difference in age (mean 53 years; min 44, max 63), CD4 count (mean 726 cells/µl [SD 168]), HIV RNA levels (mean 961 copies/ml [SD 1645]), and time-off-ART (mean 17 years; min 4.5, max 30) between the TN and NN groups (*Figure 1—figure supplement 1*). Single HIV-1 Env-specific CD19$^+$CD20$^+$IgM$^-$IgA$^-$MBCs were FACS-sorted (*Figure 1—figure supplement 2*) and we obtained repertoires of natively paired, full variable region IGH and IGL by Immune Repertoire Capture analysis; 12,591 total sequence pairs were generated for all study participants, with 5771 sequences from TN and 2707 sequences from NN (*Figure 1—figure supplement 2*). To exclude potential bias caused by the number of input cells or sequences, we normalized the proportion of each repertoire signature across all sequences in each individual (see 'Materials and methods'). TN MBCs showed an Ig subclass distribution similar to serum IgG antibodies, with average levels of IgG1 (78.6%) > IgG2 (12.5%) > IgG3 (8.8%) > IgG4 (0.1%) while Ig subclasses from NNs differed in hierarchy with average levels of IgG1 (82.5%) > IgG3 (11.1%) > IgG2 (6.2%) > IgG4 (0.2%) (*Figure 1A*).

### IGH-VDJ and IGL-VJ gene segment usage

The MBC repertoires were found to use 40 IGHV genes representing all seven human VH gene families. Of the seven IGHV gene families, IGHV1 was most commonly used in both NN and TN repertoires, consistent with prior reports that anti-HIV antibodies frequently use IGHV1 genes (*Breden et al., 2011*; *Gorny et al., 2009*; *Figure 1—figure supplement 3A*). The top two most frequently used IGHV genes were IGHV1-69 and IGHV4-34 in both the TN and NN repertoires, followed by IGHV4-4 in the TN and IGHV1-24 and IGHV1-2 in the NN repertoire (*Figure 1—figure supplement 3B*). For IGHD and IGHJ genes, no statistical significant differences were observed. The

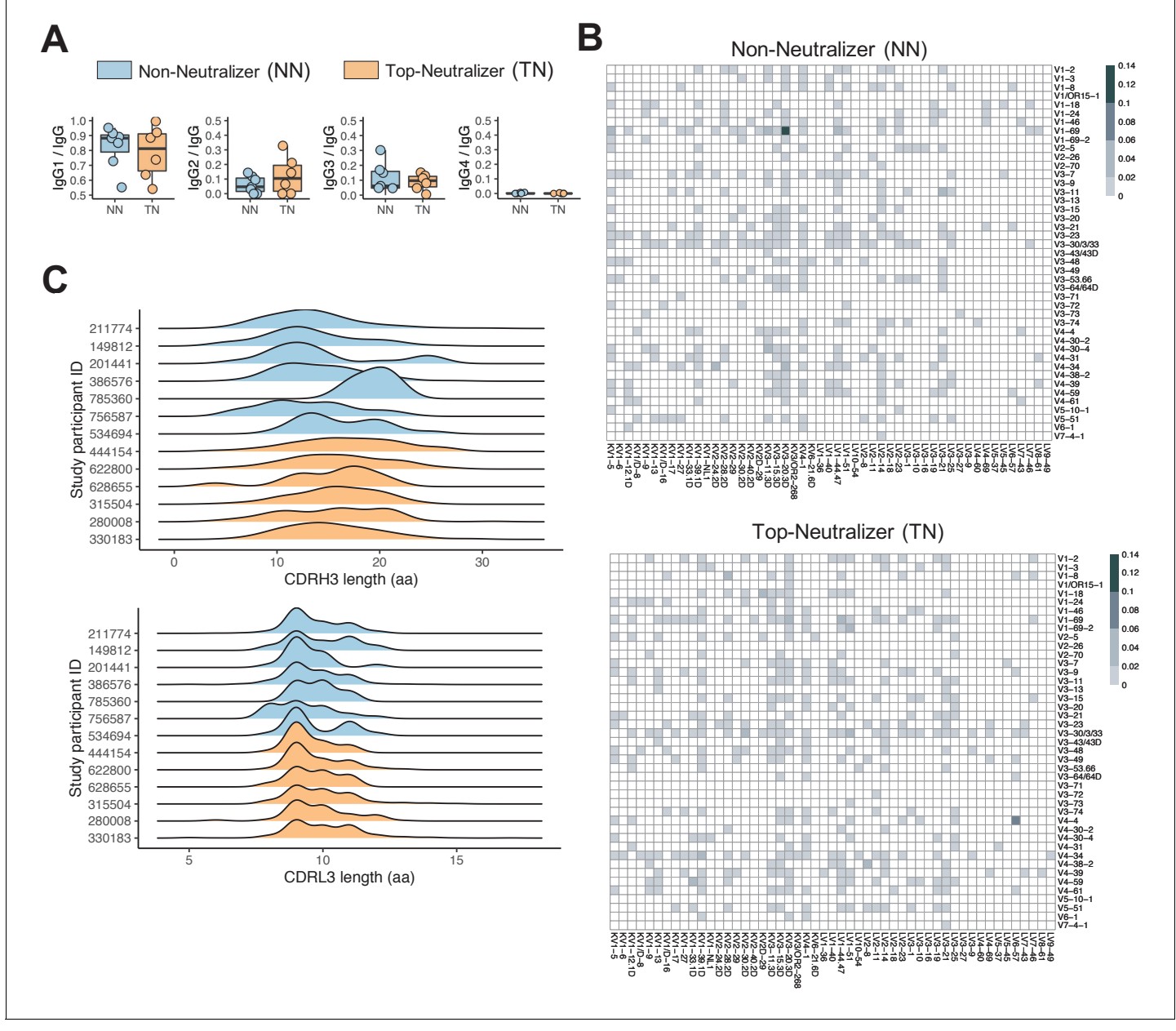

**Figure 1.** Non-neutralizer (NN) and top-neutralizer (TN) memory B-cells (MBCs) B-cell receptor (BCR) repertoires. (**A**) IgG subclass distribution in NNs and TNs. (**B**) The proportion of IGHV-IGLV gene combination in NN and TN individuals' usage was averaged and plotted in heatmaps representing the intensities of heavy and light chain paired V gene usage. (**C**) CDRH3 and CDRL3 length distribution (in aa) in NN and TN.

The online version of this article includes the following figure supplement(s) for figure 1:

**Figure supplement 1.** Non-neutralizer (NN) and top-neutralizer (TN) characteristics at time of sampling.

**Figure supplement 2.** Gating strategy to identify and sort single envelope (Env)-specific memory B cells.

**Figure supplement 3.** Non-neutralizer (NN) and top-neutralizer (TN) B-cell receptor (BCR) repertoires.

most frequently used heavy chain VDJ genes were IGHV1-69, IGHD3-22, IGHJ4 in NN and IGHV1-69, IGHD4-17, IGHJ4 in TN. Among IGLV and IGLJ genes, IGLKV3 and IGLKV3-20 dominated in both TN and NN repertoires (*Figure 1—figure supplement 3B*). This is consistent with previously reported data showing the dominance of IGLKV3-20 (*Jackson et al., 2012*).

We next compared the proportions of specific IGH/IGL pairs between TNs and NNs. IGHV1-69/IGKV3-20 combination followed by IGHV1-69/IGKV1-5 dominated in NN while IGHV4-4/IGLV6-57 followed by V1-69-2/LV1-51 were most frequently found in TNs (*Figure 1B*). Specifically the most frequent IGHV-IGHJ-IGLV-IGLJ combination in NN was IGHV1-69/IGHJ6/IGKV3-20/IGKJ5 and IGHV4-4/IGHJ6/IGLV6-57/IGLJ2 in TNs (*Figure 1—figure supplement 3C*). Overall, the frequencies of the top 3 IGHV/HJ-IGLV/LJ pairs differed significantly between NNs and TNs (Mann Whitney U-test, p<0.05). We next were interested to determine if we could find sequences of known bNAbs as listed in the CATNAP database (*Yoon et al., 2015*) and as summarized by *Sok and Burton, 2018*, using the same IGHV and IGLV gene segment usage and equal CDRH3 length as minimum criteria. Overall, very few sequences in our data matched known bNAb IGHV and IGLV gene pairs and of those none showed high levels of CDRH3 sequence similarity. No difference in these frequencies between TNs and NNs was observed. Only the V3 glycan antibody BG18 that had been identified previously in TN 622800 (*Freund et al., 2017*) was again detected in the BCR repertoire of this individual (*Supplementary file 3*). Taken together, these results suggest that V(D)J gene usage was similar overall between TNs and NNs, but the frequency and combination of IGH/IGL pairs differed between groups.

## Characteristics of CDRs

Complementarity determining regions (CDRs) play critical roles in the binding of antibodies to antigens and unusually long CDRH3s have been described for some bNAbs (*Sok and Burton, 2018*). The CDRH3/L3 length in TNs was between 3 and 37 aa and 5 and 17 aa, respectively, and 4 to 31 aa and 5 to 14 aa in the NNs, respectively. CDRH3 length distribution in TN sequences was significantly different to CDRH3 length distribution in NN sequences (p<2.2e-16 Kolmogorov-Smirnov test; *Figure 1C*). Furthermore, significant differences in length were observed for CDRH2, CDRL1 (data not shown), and CDRL3 (p<2.2e-16 Kolmogorov-Smirnov test; *Figure 1C*). There were no significant differences for CDRH1 and CDRL2. The mean CDRH3 lengths observed here are more consistent with what has been described in the CD4 binding site-specific bNAbs like VRC01 and 3BNC117, which have CDRH3 lengths of 14 and 12, respectively, while V2 and V3 glycan-specific bNAbs like PGDM1400 and PGT121 have substantially longer CDRH3 length with 34 and 26 aa, respectively (*Sok and Burton, 2018*). Nevertheless, CDRH3 length by itself is not a determining feature for neutralization potency and breadth but rather CDRH3-specific sequence characteristics that contribute to the superior binding activity of certain bNAbs. Other than for BG18, however, no CDRH3 sequences, that matched known bNAbs exactly, were indentified in our dataset.

## Rates of SHM in IGHV and IGLV

When we calculated the number of SHMs in IGHV, we noticed that the TN repertoires had significantly higher frequencies of nucleotide mutations compared to NN (mean ± SD: 54 ± 17 mutations for TN and 29 ± 8 mutations for NN, unpaired t-test, p<0.05) with more than 65% of sequences showing mutation frequencies greater than 15% (*Figure 2A*). When comparing the frequency of mutations in IGHV (TNs, NNs, and intermediate neutralizers) with the degree of serum neutralization breadth across the entire cohort, a direct correlation was observed (Spearman's rho = 0.60, p<0.01) (*Figure 2B and C*). Similarly, SHM in IGLV (*Figure 2D*) was more frequent in TN than NN (mean ± SD: 35 ± 18 mutations for TN and 14 ± 5 mutations for NN, unpaired t-test, p<0.05) with more than 50% of sequences showing mutation rates greater than 10%. Here as well, the frequencies of mutations in IGLV correlated with neutralization breadth (Spearman's rho = 0.63, p<0.01; *Figure 2E and F*). We were next interested to determine whether the IGH and IGL V-J gene usage would differ between the more (with higher than 15% mutations in IGH, 10% mutations in IGL) and less (≤15% mutations in IGH, 10% mutations in IGL) mutated sequences. While including all sequences, independent of the mutation rates, there was no apparent difference in IGHV-J (*Figure 2G*) or the IGLV-J combinations between NNs and TNs (*Figure 2H*). For the sequences with higher than 15% mutations in IGHV, however, which occurred nearly exclusively in the TNs, the pattern was distinct, demonstrating a selection for fewer dominant V-J combinations (*Figure 2G*). In particular HJ4/HV1-69-2, HJ6/HV4-4, HJ6/HV3-30.33, and HJ6/HV4-34 were significantly enriched in TN sequences with high mutation rates (p<0.01, one-sided Fisher's exact test followed by correction for multiple comparison using Benjamini-Hochberg method). The enrichment of HV4-34 usage (p<0.001, one-sided Fisher's

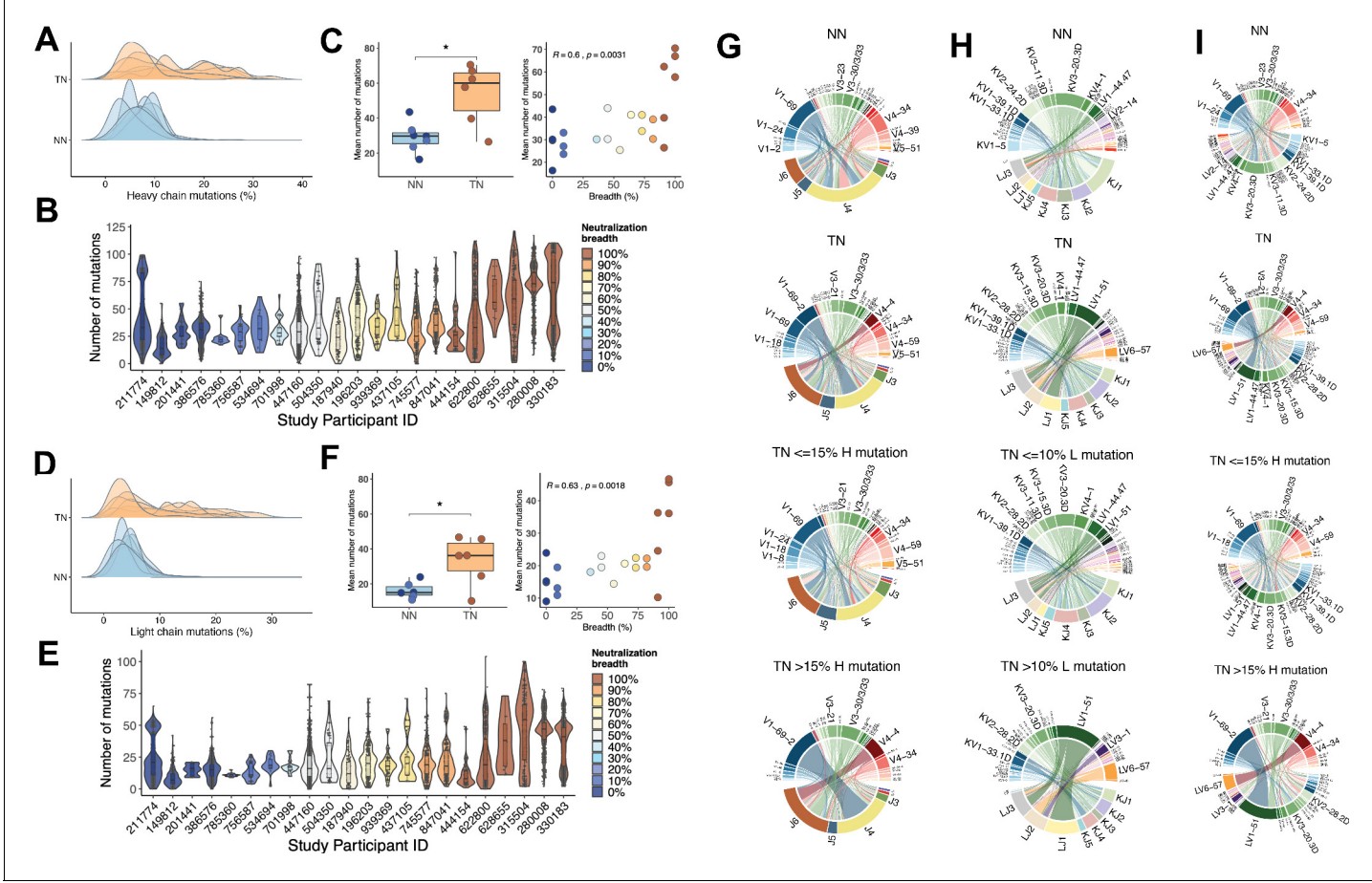

**Figure 2.** Mutation characteristics of IGHV and IGLV genes and associations with neutralization breadth. (**A**) The overall mutation frequencies in IGHV in top-neutralizers (TNs) versus non-neutralizers (NNs) are significantly different with mean number of mutations of 54 in TNs and 29 in NNs (unpaired t-test, p<0.05). (**B**) The number of mutations in IGHV for each of the 22 study participants is plotted and the serum neutralization breadth is color-coded ranging from blue (0% neutralization) to red (100% neutralization). (**C**) The mean frequency of mutations in IGHV correlates with the serum neutralization breadth in % (Spearman's rho = 0.60, p<0.01). Similarly, for IGLV, the mean number of mutations differ (35 in TNs and 14 in NNs) (**D**) and correlates with the serum neutralization breadth in % (r = 0.63, p<0.01) (**E and F**). Circos plots summarize the combinations of V and J segments used in the rearranged IGH genes (**G**) and IGL genes (**H**) and V segment combinations of IGH and IGL (**I**) expressed by memory B-cells (MBCs). Top circos graphs demonstrate combinations expressed by NN MBCs and second row graphs demonstrate TN MBCs independent of the number of detected mutations. Third and fourth row circos graphs demonstrate gene combinations expressed by TN MBCs with low and high mutation rates. For each (**G**)/(**H**) plot, the bottom half depicts J genes and the top half depicts V genes. For (**I**) top half depicts IGHV and bottom half IGLV. To exclude potential bias caused by the number of input cells, the number of sequences in each repertoire signature was weighted by the total number of sequences and total number of unique clones in each individual (see 'Materials and methods'). The arc length of each segment denotes the normalized frequency at which each gene segment was identified. Rearrangement of a J gene with a V gene segment in a clonal immunoglobulin (Ig) sequence is represented by a ribbon (ribbons carry the color of the HV or LV family of the gene participating in the pairing). The width of the ribbons corresponds to the weighted frequency at which each particular HV-HJ or LV-LJ rearrangement was used in the respective MBC repertoire.

exact test followed by correction for multiple comparison using Benjamini-Hochberg method), a feature of autoimmunity (*Wardemann et al., 2003*), in the TN sequences with high mutation rates, is consistent with a recent report that suggested that HIV infection induces a permissive state in which potentially autoreactive clones that would otherwise be eliminated can persist (*Roskin et al., 2020*). A similar contraction to fewer dominant V-J combinations was also observed for sequences with higher than 10% mutations in IGLV (*Figure 2H*). The frequencies of the paired IGHV and IGLV genes (*Figure 2I*) demonstrated a dominant pairing of the HV1-69-2 and HV3-30.33 with LV1-51, and HV4-4 with LV6-57 in TNs with >15% heavy chain mutations. Assuming that the somatic IGHV mutation rates in B-cells correlate with the total exposure time in the GC environment, and in light of the fact

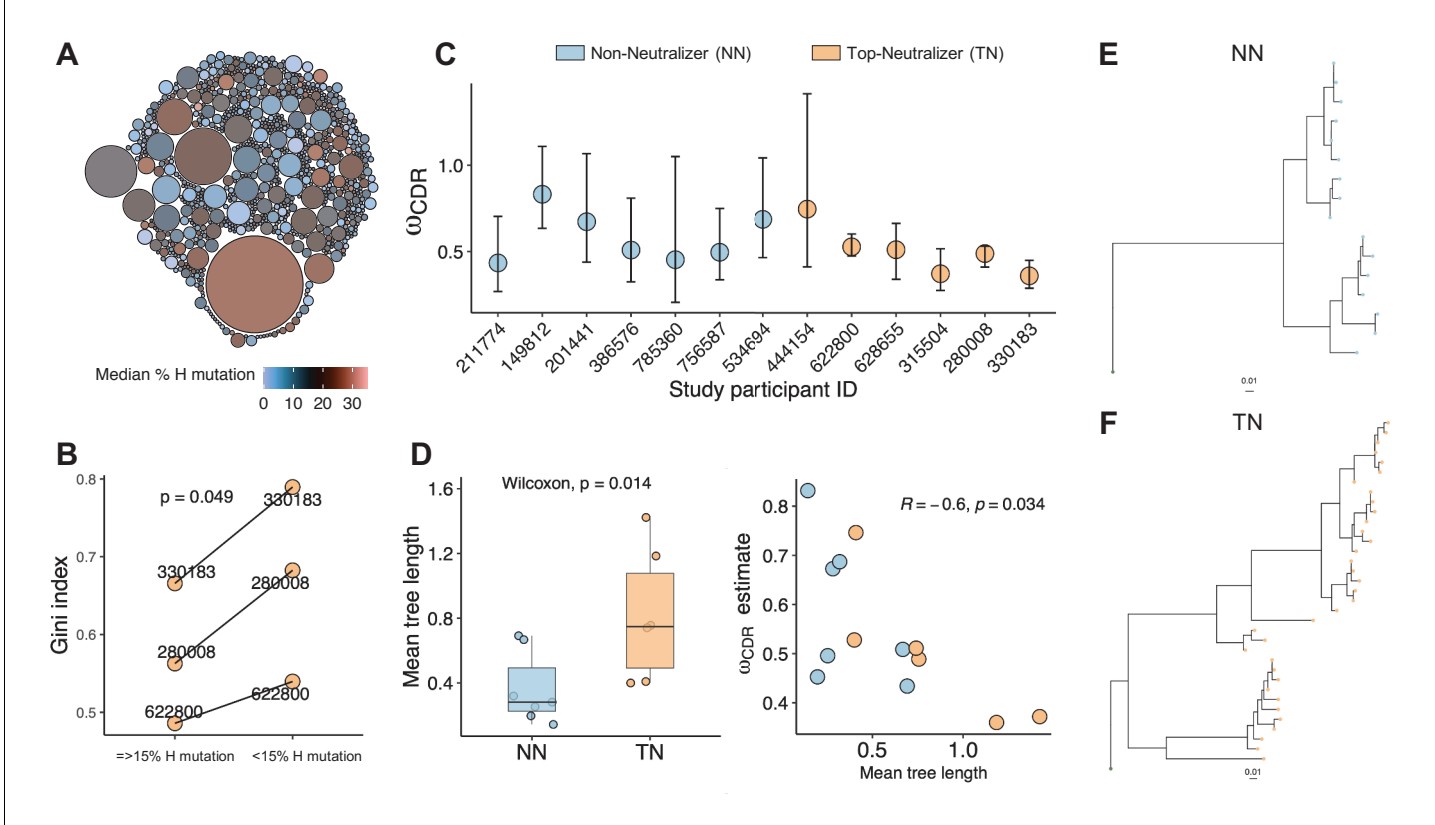

**Figure 3.** Clonality analysis. A total number of 770 clones were obtained for top-neutralizers (TNs). (A) Each clone is represented as a circle. The sizes of the circles are scaled to the number of sequences in each clone (mean number of sequences 7.4); the colors represent mean number of mutations per clone. Dominant clones consist of highly mutated sequences. (B) Gini index (0 represents perfect equality and one perfect inequality) was calculated as an estimate of clonality for TNs for which a total of more than 600 sequences were obtained (subject IDs: 330183, 280008, 622800). Per each subject, sequences were divided into two sets as high mutated sequences (>60 mutations) and low mutated sequences (≤60 mutations). Each set of sequences were clustered into clones and Gini index was calculated. Gini Index is higher (paired t-test, p=0.04) in highly mutated sequence sets supporting the presence of fewer dominant clones. (C) Maximum likelihood point estimates of ω with 95% confidence intervals. (D) Estimates of mean tree length (total substitutions per codon within a lineage, averaged across all lineages within a repertoire) were compared between TN and non-neutralizer (NN) using Wilcoxon rank sum test (left graph). The correlation between ω and mean tree length calculated using Spearman's correlation. Largest lineage trees in NNs (E) and TNs (F).

that the TN and NN did not differ in age, time-off-ART, and viral loads, our data suggests that TNs were more permissive for extensive SHM of B-cell clones.

## Clonal selection

To further assess the clonal nature of the BCR repertoire in TNs, we clustered sequences into clonal groups. Each unique clone is a cluster of sequences that meet the following criteria: (1) derived from the same individual; (2) share the same V and J gene segment annotations in heavy chain and light chain; (3) have equal CDRH3 length; and (4) junction regions share similarity by a defined sequence distance cutoff (see 'Materials and methods'). This clonal assignment identified 770 clones (mean number of sequences 7.4) and showed that larger sized clusters (dominant clones) consist of highly mutated sequences (*Figure 3A*). To quantify the greater clonality of the highly mutated sequences, we used an unevenness measure, the Gini index. We selected individuals (subject IDs: 330183, 280008, 622800) for whom a total of more than 600 sequences were available. Sequences were divided into two sets, highly mutated (>15% mutations) and less mutated (≤15% mutations) sequences. Each set of sequences was clustered into clones and the Gini index was calculated based on the clone size distributions. For all three

subjects, the Gini index was greater in the highly mutated set (paired t-test, p<0.04), thus supporting the observation that the BCR repertoire becomes more narrow with increasing mutation rates, suggesting the dominance of certain clones (*Figure 3B*).

Next, we estimated the evolutionary process of mutation and selection in B-cells using phylogenetic models. Selective dynamics have been estimated using ω: dN/dS, the ratio of nonsynonymous (amino acid replacement) and synonymous (silent) mutation rates (*Nielsen and Yang, 1998*; *Hoehn et al., 2019*). Low $\omega_{CDR}$ values indicate fewer amino acid changes in CDRs than expected suggesting negative BCR selection to remove affinity-decreasing variants, while positive selection to introduce new affinity-increasing amino acid variants is associated with higher $\omega_{CDR}$ values. When comparing TNs to NNs we found an overall trend toward fewer amino acid changes in the CDRs of TNs (maximum likelihood estimate of $\omega_{CDR}$ in *Figure 3C*) indicating stronger purifying selection. Furthermore, the average lineage tree lengths ( the total expected substitutions per codon site within an individual lineage phylogeny) was significantly higher in TNs compared to NNs (p<0.05, Wilcoxon rank sum test) and correlated negatively with ω (*Figure 3D*). The largest lineage trees constructed in TNs and NNs are visualized in *Figure 3E* and *Figure 3F*, respectively. These results are therefore consistent with prior observations that B-cell lineages shift toward negative selection over time as a general feature of affinity maturation (*Hoehn et al., 2019*) and suggest that the dominant clones in the TNs have reached a level, where the continuous accumulation of additional random amino acid changes would rather be detrimental and decrease affinity (*Clarke et al., 1985*).

## Discussion

Overall, when comparing frequencies of either IGH or IGL genes between those individuals who developed antibody neutralization breadth compared to those who did not, the repertoire analysis did not reveal substantial differences. This is consistent with the findings from Scheepers et al. who analyzed the IGHV repertoires of bulk peripheral blood B-cells for 28 HIV-1 clade C infected South African women with different ranges of serum neutralization breadth (*Scheepers et al., 2015*) and observed a wide range in the number of IGHV alleles in each individual but did not find significant differences in germline IGHV repertoires between individuals with or without bNAbs. When we however compared the repertoire of paired IGH/IGL sequences between both groups, clear differences in the hierarchy of IGHV/IGHJ/IGLV/IGLJ combinations were observed. Despite these distinguishing pattern, we did not observe an enrichment of known bNAb sequences in the TNs. In fact, we found very few sequences in our cohort that matched IGHV-IGLV gene combinations and/or CDRH3 - sequences of published Nabs (*Sok and Burton, 2018*) with BG18 being the exception. Indeed, additional bNAbs, such as the CD4 binding site antibody 3BNC117, have been isolated from HIV controllers (*Scheid et al., 2011*). The general absence of these or similar bNAb sequences in our data therefore suggests that the neutralization breadth observed in the TNs was either from single bNAbs with non-classical epitope specificities or due to multiple neutralizing antibodies targeting multiple different epitopes. This distinction necessitates the generation of antibodies from a large number of BCRs in the TNs for characterization of their neutralizing activities and such studies are planned in the future.

In addition to the composition of the BCR repertoire, that separated TNs from NNs, both groups were clearly distinguishable by the significantly higher rates of mutations in the VH and VL genes in the individuals with neutralizing antibody breadth. Our data in HIV-1 specific B-cells confirms findings recently reported by Roskin et al. in a bulk B-cell repertoire analysis, demonstrating that individuals with neutralizing antibody breadth have hundreds to thousands of antibody lineages with long CDRH3s and very high SHM frequencies (*Roskin et al., 2020*). Moreover, the mutation frequency in our data correlated directly with the neutralization breadth consistent with the notion that high levels of SHM are generally necessary for neutralization breadth and potency (*Klein et al., 2013*).

Initial interclonal and then intraclonal competition between B-cells during affinity maturation is well described (*Kuraoka et al., 2016*; *Tas et al., 2016*), resulting in diverse repertoires. Indeed, we found that with increasing numbers of mutations in HV, fewer clones with less diverse HV-HJ combinations dominated, suggesting a process that allows these clones to mature and accumulate mutations. Since GCs select for affinity and not for neutralizing breadth and since evolving viruses, such as HIV-1, will continuously elicit novel immune responses from naive B-cells, strong effective

selection normally occurs, so that antibodies specific to the contemporaneous virus have a sufficient advantage to outcompete any less-specific potential broad neutralizers before they take hold. Indeed, by focusing on the kinship relations among Ig gene segment recombinations within individual clones, we found evidence of natural selection in the NNs, with younger and less diverse clones continuously evolving, constantly exploring novel evolutionary space. Conversely, in TNs, this continuous evolution does not seem to occur, but rather specific evolutionary paths are pursued. One potential mechanism behind this observation is that neutralizers may generate massive bursts of clones that compete aggressively and prevent the evolution of lower affinity variants. Alternatively, the effects of immunodominance where B-cell specific to an epitopic site dominate B-cells that target other sites, as has been described in influenza infection (*Angeletti et al., 2017*), might not occur to the same degree in TNs as it might do in NNs. Indeed, a generalized deficiency of normal selection against B-cells expressing IgG with long CDRH3 regions and high SHM frequencies has been associated with the development of HIV neutralizing breadth (*Roskin et al., 2020*).

While the selection process might be suppressed in the TNs early on, to allow extended SHM with ongoing clonal evolution and increasing affinity, this process seems to revert during the later stages of antibody evolution when the benefit of additional amino acid changes decreases, and purifying selection becomes dominant and removes most non-synonymous mutations. Indeed, Yaari et al. demonstrated that early mutations in the trunk of B-cell clonal lineages from healthy subjects were consistent with positive selection pressure than more recent mutations in the canopy (*Yaari et al., 2015*) and a negative relationship between both signs of increased negative selection (low ωCDR values) and mean repertoire tree length have been reported across subjects of different age and sex (*Hoehn et al., 2019*). Interestingly, Wu and Sheng et al. observed that the evolutionary rates of HIV-1 bNAb antibody lineages derived from different donors consistently decreased over time (*NISC Comparative Sequencing Program et al., 2015*; *Sheng et al., 2016*), also suggesting that SHM occurs at a faster rate during the early phase of lineage development while later on, a decrease in mutation rates may be a mechanism to protect immune memory.

Several potential mechanisms for differences in BCR evolution and selection can be envisioned in TNs versus NNs. B-cells with the highest affinity receptors tend to acquire the most antigen from the follicular dendritic cell network and present the highest density of cognate peptides to CD4+ T-follicular helper (Tfh) cells, which respond with survival signals to the B-cell. While some studies have suggested that the quality and quantity of GC Env-specific Tfh cells are associated with the expansion of Env-specific B-cells and broader antibody neutralization activity (*Yamamoto et al., 2015*; *Moody et al., 2016*), others have failed to find a correlation (*Roskin et al., 2020*). Interestingly, Cirelli et al. recently demonstrated that slow delivery of antigen resulted in enhanced autologous tier 2 nAb development in non-human primates by inducing higher frequencies of total and Env-specific GC-Tfh cells accompanied by larger and more diverse Env-specific B-cell lineages. In this model, evolving B-cells with BCRs able to bind to a large array of Env variants, are likely able to capture more antigen, present more effectively to Tfh cells, and thus gain higher survival signals. Conversely, B-cells able to bind tightly, but to only a narrow range of Env variants, will only capture a fraction of arrayed antigens and receive less help. Thus, high affinity, narrow specificities may gain less support compared to lower -affinity, broad -binders (*De Boer and Perelson, 2017*). Thus, while antigenemia is lower in controllers, the low-level titration of antigen into the system may create a higher bar of selection for evolving B-cell clones. Along with this high degree of competition, B-cell lineages in HIV controllers able to bind many variants may capture larger amounts of antigen and therefore gain enhanced signals from Tfh cells (*Fukazawa et al., 2015*). Additionally, HIV controllers with low-level viremia also exhibit unique inflammatory profiles, and sustained GC activity, collectively also potentially providing an optimal environment for B- cell development.

In summary, development of B-cell lineages with superior neutralization breadth in HIV controllers is linked to selected clonal evolution, therefore allowing B-cells to accumulate mutations and to diversify without the constant risk of clonal interference. Immunization strategies that facilitate this process by providing continuous, diverse, and subtherapeutically dosed antigen, in the setting of target inflammatory signals, could potentially accelerate the development of broad nAb responses in vaccinees.

# Materials and methods

## Key resources table

| Reagent type (species) or resource | Designation | Source or reference | Identifiers | Additional information |
|---|---|---|---|---|
| Chemical compound, drug | Propidium iodide | Life Technologies | Cat# P1304MP | |
| Antibody | Anti-CD3 (FITC, clone UCHT1) | Biolegend | Cat# 300406, RRID:AB_314060 | 10 µg/ml |
| Antibody | Anti-CD14 (FITC, clone HCD14) | Biolegend | Cat# 325604, RRID:AB_830677 | Dilution (1:100) |
| Antibody | Anti-IgM (FITC, clone MHM-88) | Biolegend | Cat# 314506, RRID:AB_493009 | Dilution (1:100) |
| Antibody | Anti-CD20 (PE-cy7, clone 2H7) | BD Biosciences | Cat# 560735, RRID:AB_1727450 | Dilution (1:100) |
| Antibody | Anti-CD19 (BV421, clone HIB19) | Biolegend | Cat# 302233, RRID:AB_10897802 | Dilution (1:100) |
| Antibody | Anti-IgA (FITC, clone IS11-8E10) | Miltenyi Biotec | Cat# 130-093-071, RRID:AB_1036156 | Dilution (1:100) |
| Antibody | hCD40L His tag | BPS Biosciences | Cat# 71191 | Dilution (1:100) |
| Chemical compound, drug | Streptavidin conjugated to PE or AlexaFluor 647 | Life Technologies | | |
| Recombinant protein | Clade B JR-CSF gp120, JR-FL gp140, 92BR020 gp120, clade A BG505 SOSIP, clade C IAVI C22 gp120 | Duke Human Vaccine Institute, protein production facility | | |
| Cell line (*Homo sapiens*) | TZM-bl cells | NIH AIDS Reagent Program | Cat# 8129–442, RRID:CVCL_B478 | |
| Recombinant DNA reagent | AC10.0.29, RHPA4259.7, THRO4156.18, REJO4541.67, WITO4160.33, TRO.11, SC422661.8, QH0692.42, CAAN5342.A2, PVO.4 TRJO4551.58 | Seaman lab | | Plasmids |
| Commercial assay or kit | Ampure XP beads | Beckman Coulter | Cat# A63881 | |
| Commercial assay or kit | Quant-iT PicoGreen dsDNA Assay Kit | Invitrogen | Cat# P7589 | |
| Software, algorithm | FlowJo | https://www.flowjo.com/solutions/flowjo | RRID:SCR_008520 | |
| Software, algorithm | IgBLAST | https://www.ncbi.nlm.nih.gov/igblast/ | RRID:SCR_002873 | |
| Software, algorithm | 454 GS FLX data analysis software | Roche | RRID:SCR_018028 | |
| Software, algorithm | R version 4.0.0 | https://cloud.r-project.org | RRID:SCR_001905 | |
| Software, algorithm | Igphyml (version 1.1.0) | https://igphyml.readthedocs.io/en/latest/ | | |
| Software, algorithm | Alakazam (version 1.0.0) | https://alakazam.readthedocs.io/en/1.0.1/news/ | | |

## Human subjects

A total of 22 HIV-infected individuals were recruited for this study. All participants underwent leukapheresis and routine blood draws as per study protocol. All subjects signed informed consent, and the study was approved by the MGH/Partners Institutional Review Board (Protocol # 2003P001894).

## Neutralization assay

HIV-1 neutralization breadth was assessed using the Tzm-bl cell-based pseudovirus neutralization assay, as described (*Sarzotti-Kelsoe et al., 2014*), against a standard panel of Env-pseudoviruses derived from nine clade B tier 2 viruses: AC10.0.29, RHPA4259.7, THRO4156.18, REJO4541.67, WITO4160.33, TRO.11, SC422661.8, QH0692.42, CAAN5342.A2, and two tier 3 viruses: PVO.4 and TRJO4551.58 (*Li et al., 2005*). Murine leukemia virus (MuLV) was included in all assays as a negative control. Neutralization titers (50% inhibitory dose [ID50]) were defined as the reciprocal of the plasma sample dilution that caused a 50% reduction in relative luminescence units (RLUs) compared to virus control wells after subtraction of background RLUs. Neutralization breadth was determined as the proportion of pseudoviruses with an ID50 score threefold above background titers observed against MuLV negative control virus (3× ID50 of MuLV) (*Sarzotti-Kelsoe et al., 2014*).

## Cell lines

TZM-bl cells (also called JC53BL-13) were obtained from the NIH AIDS Reagent Program and authenticated by (1) morphology and growth characteristics, (2) cell surface expression of CD4 and co-receptors was assessed by flow cytometry, and (3) susceptibility in vitro to HIV, with readout being firefly Luc activity in infected cell lysates. All cell lines were tested to confirm absence of myco-plasma contamination.

## Single-cell flow cytometry sorting

Isolated cells were stained with fluorochrome-antibody conjugates and reagents to identify antigen-specific MBCs. The panel consisted of propidium iodide (Life Technologies); CD3 (FITC, clone UCHT1), CD14 (FITC, clone HCD14), IgM (FITC, clone MHM-88), CD20 (PE-cy7, clone 2H7), CD19 (BV421, clone HIB19) (All Biolegend); IgA (FITC, clone IS11-8E10) (Miltenyi Biotec). Preformed conjugates for antigen-specific B-cell sorting were made as described (*Sok et al., 2014*) using streptavidin conjugated to PE or AlexaFluor 647 (Life Technologies). B-cell probes were made using clade B JR-CSF gp120, JR-FL gp140, 92BR020 gp120, clade A BG505 SOSIP, and clade C IAVI C22 gp120 tags (Duke Human Vaccine Institute, protein production facility) and a cocktail of all antigens was used for cell sorting. IgG+ B-cells were defined as CD3/14−, CD19+, CD20+, and IgA/IgM-; antigen-specific B-cells positive for probes in either PE or APC color were sorted into 96-well U-bottom plates containing 200 µl of B-cell culture medium (IMDM supplemented with FBS, Normocin, hIL-2, hIL-21, hCD40L His tag, and anti-His antibody). After 4-day culture, the B-cells were sorted into microtiter plates at one cell per well. Sorted plates were frozen immediately and maintained at −80℃ before reverse transcription (RT)/PCR.

## BCR sequencing

Natively paired variable region sequences from individual cells were generated by RT, cDNA barcoding, amplification, and sequencing as described previously (*Tan et al., 2014*; *DeFalco et al., 2018*). cDNA sequences were determined by 454 Titanium sequencing. A minimum of 10 reads for each chain (heavy and light) was required, and a contig was kept only if it included at least 90% of the reads for that chain from that well. V(D)J assignment and mutation identification was performed using IMGT reference sequences and IgBLAST through Immcantation analysis framework (*Supplementary file 2*). Percentage SHM was calculated by calling observed mutations of the entire input sequence compared to the germline sequence.

## Analysis

Statistical analyses and visualizations were performed in R version 4.0.0 (April 24, 2020). Subclass distribution, gene family, and gene usage analysis: to exclude potential bias caused by the number of input cells or sequences, we divided the number of occurrences of each repertoire signature by the total number of all sequences for each individual. Gene recombination profiles: to avoid skewing of mean frequencies when averaged across all individuals by individuals with high or low number of available sequences (in particular for sequences with higher than 15% heavy chain mutations), normalized frequencies were multiplied by a factor (number of unique clones per individual). For IGHV-IGLV recombinations, the highly mutated sequences were defined using only heavy chain mutation

cutoff. CDRH3 length distributions consist of CDRH3 length of each unique clone. Kolmogorov-Smirnov test was used to compare the distributions between TN and NN repertoires.

Mean number of mutations between TN and NN were compared using unpaired two-samples t-test. All correlations were performed using Spearman's correlation. Visualizations were performed using R package ggplot2 (version 3.3.0). The flow graph of IGHV/IGHJ/IGLV/IGLJ combinations were plotted using ggalluvial extension of ggplot2. Circos plots were generated using R package circlize (version 0.4.9) *chordDiagram* function. The layout for clonal network visualization was arranged using R package packcircles (version 0.3.3).

Clones were defined using DefineClones tool of the Immcantation analysis framework (50). Clones consist of sequences that meet the following criteria: (1) come from the same individual; (2) share the same IGHV and IGHJ gene segment annotations; (3) have equal CDRH3 length; and (4) share sequence similarity calculated by nucleotide Hamming distance threshold of 0.16. Clonal groups were then split that have differing IGLV and IGLJ gene segment annotations.

The clonality measure was estimated using Gini index and calculated using R package bcRep (version 1.3.6) *clones.giniIndex* function.

Lineage trees were constructed using IgPhyML (version 1.1.0) (*Stern et al., 2014*). The HLP19 model was used to estimate maximum likelihood tree topologies. Clonal selection was estimated using $\omega$, also called dN/dS, or the ratio of nonsynonymous (amino acid replacement) and synonymous (silent) mutation rates. A value of $\omega \approx 1$ indicates totally neutral amino acid evolution, $\omega < 1$ negative selection, and $\omega > 1$ diversifying selection. The topology analyses were carried out using Alakazam (version 1.0.0) (*Gupta et al., 2015*) and the lineage trees were visualized using R ggtree (version 2.2.1).

# Acknowledgements

We would like to thank the participants of the Ragon HIV controller studies from whom these samples were obtained, and the staff involved in all the sample collection and processing. We thank Dr Daniel Lingwood for the careful review of the manuscript and his critical input. This work was funded by Bill and Melinda Gates Foundation Collaboration for AIDS Vaccine Discovery (CAVD) grant #OPP1066973 and #OPP1146996.

# Additional information

## Competing interests

Dongkyoon Kim, Guy Cavet, Ngan Nguyen: employee of Atreca Inc. Yann Chong Tan: Former employee of Atreca Inc Now employee of Esco Ventures X. Galit Alter: GA is a founder of Seromyx Systems. The other authors declare that no competing interests exist.

## Funding

| Funder | Grant reference number | Author |
| --- | --- | --- |
| Bill and Melinda Gates Foundation | OPP1146996 | Michael S Seaman |
| Bill and Melinda Gates Foundation | OPP1066973 | Galit Alter |
| Ragon Institute of MGH, MIT and Harvard | Internal funds | Boris Juelg |

The funders had no role in study design, data collection and interpretation, or the decision to submit the work for publication.

## Author contributions

Deniz Cizmeci, Data curation, Formal analysis, Validation, Visualization, Methodology, Writing - original draft, Writing - review and editing; Giuseppe Lofano, Formal analysis, Visualization, Methodology, Writing - review and editing; Evan Rossignol, Formal analysis; Anne-Sophie Dugast,

Conceptualization, Resources, Data curation, Methodology, Writing - review and editing; Dong-kyoon Kim, Data curation, Formal analysis, Validation, Writing - review and editing; Guy Cavet, Data curation, Formal analysis, Supervision, Validation, Methodology, Writing - review and editing; Ngan Nguyen, Yann Chong Tan, Data curation, Formal analysis, Validation, Methodology, Writing - review and editing; Michael S Seaman, Data curation, Formal analysis, Funding acquisition, Validation, Methodology, Writing - review and editing; Galit Alter, Conceptualization, Supervision, Funding acquisition, Investigation, Project administration, Writing - review and editing; Boris Julg, Conceptualization, Supervision, Investigation, Visualization, Writing - original draft, Writing - review and editing

#### Author ORCIDs
Boris Julg  https://orcid.org/0000-0002-4687-9626

#### Ethics
Human subjects: All subjects signed informed consent, and the study was approved by the MGH/Partners Institutional Review Board (Protocol #: 2003P001894).

#### Decision letter and Author response
Decision letter https://doi.org/10.7554/eLife.62648.sa1
Author response https://doi.org/10.7554/eLife.62648.sa2

## Additional files

#### Supplementary files
• Supplementary file 1. Study participant characteristics. CD4+ T-cell counts (cells/μl), viral loads (HIV RNA copies/ml) and neutralization titers (50% inhibitory dose [ID50], 1/x] from each individual against a panel of 11 tier 2 and 3 envelope (Env)-pseudoviruses and murine leukemia virus (MuLV)-pseudotyped virion control. The right column (breadth %) shows the % of viruses that are neutralized above background (3× MuLV control). Six individuals with neutralization of more than 90% of the panel classified as top-neutralizer (TN) while neutralization of less than 10% of the panel viruses classified as non-neutralizer (NN).

• Supplementary file 2. Annotated sequences.

• Supplementary file 3. Sequences similarity with known broadly neutralizing antibody (bNAb) families.

• Transparent reporting form

#### Data availability
All data generated or analysed during this study are included in the manuscript and supporting files. A source data file (Supplementary file 2) has been provided for Figures 1-3.

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
