## [Decision Letter]

**Acceptance summary:**

This study clearly shows that the donor antibodies' levels of somatic hypermutation correlate with serum neutralization breadth. The authors suggest that B cells expressing BCRs capable of binding a large array of Env variants gain more T help than B cells expressing BCRs that target a narrower range of Env variants, which is an interesting hypothesis for interpreting immune responses in the context of a continuously-evolving virus.

**Decision letter after peer review:**

Thank you for submitting your article "Distinct clonal evolution of B-cells in HIV controllers with neutralizing antibody breadth" for consideration by *eLife*. Your article has been reviewed by 3 peer reviewers, one of whom is a member of our Board of Reviewing Editors, and the evaluation has been overseen by Satyajit Rath as the Senior Editor. The reviewers have opted to remain anonymous.

The reviewers have discussed the reviews with one another and the Reviewing Editor has drafted this decision to help you prepare a revised submission.

Our expectation is that the authors will eventually carry out any additional experiments, if necessary, and report on how they affect the relevant conclusions either in a preprint on bioRxiv or medRxiv, or if appropriate, as a Research Advance in *eLife*, either of which would be linked to the original paper.

Summary:

Cizmeci and colleagues determined the serum neutralizing antibody activities of 22 HIV-infected individuals that spontaneously controlled HIV replication to low plasma levels in the absence of therapeutic interventions. They identified six individuals whose plasmas displayed broad neutralizing activities (top neutralizers, TN) and seven with narrow/no neutralizing activities (non-neutralizers, NN). They then isolated Env-specific class-switched memory B cells from these individuals and sequenced their IGH and IGL genes. This allowed them to determine the frequencies of specific IGH and IGL gene segments (including germline genes of known broadly neutralizing antibodies) and to compare these frequencies between TNs and NNs. Their analysis clearly shows that the donor antibodies' levels of somatic hypermutation correlate with serum neutralization breadth. The authors propose that in the context of prolonged, but relatively low HIV replication, B cells expressing BCRs capable of binding a large array of Env variants may gain more T help than B cells expressing BCRs that target a narrower range of Env variants. The former BCRs could be those from which broadly neutralizing antibodies are derived from, while the later ones could be those that give rise to neutralizing antibodies with narrow breadth. In the context of a continuously evolving virus, this proposal makes sense. The data presented in this manuscript does provide a valuable view into the natural development of neutralization breadth. However, the reviewers have substantial concerns, as outlined below, that they invite you to address in a revised manuscript. While they appreciate the valuable dataset presented by the authors, they believe that the analysis could be more thorough, and that further exploration of neutralizing Abs from the TN would strengthen the study.

Essential revisions:

1. The authors report that the composition of the BCR repertoire alone was insufficient to identify the TNs, although higher rates of somatic mutations were observed in the TNs than NNs. It is questionable whether they can draw this conclusion from the way the sequence analysis was performed. Although it is mentioned (first line in the Discussion) that 'paired' IGH and IGL sequences were compared, what is discussed throughout the manuscript are frequencies of either IGH or IGL but not frequencies of specific IGH/IGL pairs. For example, they mention that 'certain genes that are distinctive for known bNAbs, like VH1-2 (and they discuss VRC01-class antibodies)…..'. However, VRC01-class antibodies specifically utilize the *02 VH1-2 allele. Also, VRC01-class antibodies are formed by the pairing of VH1-2*02 with LCs expressing 5 amino acid long CDRL3 domains. We expect that most human VH1-2*02 HCs are paired with LCs expressing >5 amino acid long CDRL3s. They also mention that the CDRH3 lengths were not different between the IGHs found in the TN and NN (and they discuss that certain bnAbs that bind the apex of the trimer on the N332 site). On its own, the length of the CDRH3 loop is unlikely to be a good predictor of antibody neutralizing potency or breadth, because specific amino acids sequences CDRH3 loops of particular lengths are linked with the mechanism by which some anti-apex or anti-N332 antibodies act. The authors could determine the relative frequencies of VRC01-class BCRs from the existing data. Similarly, they could mine their data to identify particular CDRH3 sequences. This would be the minimal they could do prove their point, which would actually necessitate the generation of antibodies from particular BCRs and the characterization of their neutralizing activities.

2. It is unclear whether the broad neutralizing activities in the samples examined here targeted a single or multiple envelope epitopes. The authors should explicitly discuss this point in the Discussion section.

3. Several parts of the discussion hard to read and confusing. This reflects the somewhat confusing and contradictory findings of high levels of SHM, but contracting evolution that is shown in Figure 3. The discussion of this could do with some tightening. Parts are speculative and wordy, e.g., the first paragraph of page 8 could be shortened.

4. It is unclear whether a cocktail of the five different Envs listed in the Methods section was used to sort B cells, and if the same Envs were used in every case or not.

5. The manuscript reports that the difference in CDRH3 length distribution (Figure 1F) is not statistically significant (they do not report the exact p value produced by the Wilcoxon rank sum test). However, the difference appears substantial in Figure 1F. One wonders whether a different statistical test or looking at subpopulations might give a statistically-significant result. Also, in Figure 1F, can the authors comment on the very high frequency of sequences in NN with CDRH3 lengths of 9 and those of 16 on the TN? Are these sequence artefacts, perhaps of oversampled clones in certain donors?

6. The authors state that sequence data will be deposited in GEO. The processed sequences (used to make Figure 2, etc.) should be made available, not just the raw NGS sequencing.

7. There is some inconsistency with the V gene names – "VH1-2" vs. "HV4-59".

8. The proposal in the Introduction that neutralizing breadth may be achievable in the absence of high levels of viremia is not entirely new, as broadly neutralizing antibodies have been isolated from elite controllers. For instance, Kwong and colleagues reported on the presence of broadly neutralizing antibody responses in a cohort of long-term non-progressors (Gonzalez et al. PLOS ONE 2018) and the Bjorkman group reported on the isolation of a broadly neutralizing monoclonal antibody from a 'controller' (Schoofs et al. Immunity 2019).

9. Page 4 – The authors discuss the varying Ig subclass distributions and suggest for example that the higher percentage of IgG3 in NN versus TN might suggest a higher proportion of more recently induced MBCs in the NNs. These differences are not very convincing, if anything they are driven by a single data point, and we suggest removing related discussion.

10. Figure 1F – Can the authors comment on the very high frequency of sequences in NN with CDRH3 lengths of 9 and those of 16 on the TN. Are these sequence artefacts, perhaps of oversampled clones in certain donors?

11. Page 5 – could the authors translate frequency of nucleotide mutations into an approximate % SHM for easy comparison to other studies particularly of bNAbs. How close are these mutation counts of >60 to the levels of SHM we associate with neutralization breadth?

12. Figure 2 – In both panels A (IGHV) and C (IGLV), while the NNs largely cluster towards the left (with low numbers of mutations), there are a cluster of NN sequences with higher numbers of mutations (~80 for IGHV and 50 for IGLV). Are these sequences from specific donors? What, if anything, distinguishes them from the other sequences?

13. Page 6 – the last statement that the first paragraph seems somehow circular – "Assuming that the somatic IGHV mutation rates in B cells correlate with the total exposure time in the GC environment, and in light of the fact that the TN and NN did not differ in age, time-off-ART and viral loads, our data suggests that in the TNs, B-cell clones were selected for extensive SHM." How would B cell clones be selected FOR extensive SHM?

14. Page 6 – Clones may show variation in terms of CDRH3 length – did the authors explore the effect of softening these criteria e.g. by allowing +/- 1 amino acid on clone size?

15. Figure 3 – legend refers to a mean number of sequences per clone of 7.7 that differs from the text.

16. Page 7, third line, first para of discussion – why "in contrast"?

"Moreover, the mutation frequency in our data correlated directly with the neutralization breadth suggesting a skewing towards poly- or cross-reactive antibodies that are able to bind mutated forms of the original epitope" – do high mutation frequencies necessarily translate to polyreactivity?

17. In the Results section, at the end of clonal selection section (page 7), the discussion of "maximum affinity maturation" does not seem to reckon with the continuous change in the Env antigen, which generally manages to escape antibody pressure.

18. Page 8 – correct spelling of "alternatively"

19. Page 2 and page 20 – Supplemental Table 1. Why have the authors used a background of 3x MuLV? This is different from most studies of neutralization breadth. Why do some donors have activity against MuLV? What contributes to this non-specific background? Also, the legend contains a typo – please correct.

---

## [Author Response]

Essential revisions:1. The authors report that the composition of the BCR repertoire alone was insufficient to identify the TNs, although higher rates of somatic mutations were observed in the TNs than NNs. It is questionable whether they can draw this conclusion from the way the sequence analysis was performed. Although it is mentioned (first line in the Discussion) that 'paired' IGH and IGL sequences were compared, what is discussed throughout the manuscript are frequencies of either IGH or IGL but not frequencies of specific IGH/IGL pairs.

We appreciate the comment and have modified the data presentation and result section to highlight specifically IGH and IGL pairs. While we believe that reporting the frequencies of each IGH or IGL genes independently is informative (now in supplemental Figure S3B), we focus for the comparison of TN and NNs on IGH/IGL pairs depicting IGHV-IGLV pair frequencies as a heatmap (Figure 1B) but also listing the most frequent IGHV-IGHJ-IGLV-IGLJ combinations (in NN: IGHV1-69/IGHJ6/IGKV3-20/IGKJ5, and in TN: IGHV4-4/IGHJ6/IGLV6-57/IGLJ2) (Figure 1—figure supplement 3C). For the clonal analysis in figure 3, we only utilized the paired nature of the dataset by grouping sequences into unique clones only if they have the same IGHV-IGHJ-IGLV-IGLJ genes.

For example, they mention that 'certain genes that are distinctive for known bNAbs, like VH1-2 (and they discuss VRC01-class antibodies)…..'. However, VRC01-class antibodies specifically utilize the *02 VH1-2 allele. Also, VRC01-class antibodies are formed by the pairing of VH1-2*02 with LCs expressing 5 amino acid long CDRL3 domains. We expect that most human VH1-2*02 HCs are paired with LCs expressing >5 amino acid long CDRL3s. They also mention that the CDRH3 lengths were not different between the IGHs found in the TN and NN (and they discuss that certain bnAbs that bind the apex of the trimer on the N332 site). On its own, the length of the CDRH3 loop is unlikely to be a good predictor of antibody neutralizing potency or breadth, because specific amino acids sequences CDRH3 loops of particular lengths are linked with the mechanism by which some anti-apex or anti-N332 antibodies act. The authors could determine the relative frequencies of VRC01-class BCRs from the existing data. Similarly, they could mine their data to identify particular CDRH3 sequences. This would be the minimal they could do prove their point, which would actually necessitate the generation of antibodies from particular BCRs and the characterization of their neutralizing activities.

As suggested by the reviewer, we mined the data set for the frequencies of known bNAbs, including the VRC01-class antibodies. For the latter, we identified 3 unique clones utilizing IGHV1-2/IGKV3-20 genes, however these antibodies had 8-9 aa long CDRL3 and no sequence similarity for CDRH3. We screened for all neutralizing antibodies listed in the comprehensive review by Sok and Burton, Nat Immunology 2018 and also compared our sequences to Nabs that are listed in the CATNAP database.

We found very few sequences that match IGHV-IGLV genes with known bNAbs in either TN or NNs and overall there was no difference (Supplementary file 3). Beside the V3 glycan antibody BG18, that had previously been identified in one of our TNs (622800), we did not find any sequences that matched the exact CDR3-sequences of known bNAbs. However, these results are not surprising given that few of the TN subjects exhibited classic epitope-specific reactivity. Specifically, 22 controllers were profiled against a panel of epitope specific knock out variants in collaboration with Lynn Morris (Author response image 1) (Approach described in PMID: 24204277, PMID: 29950423). 5 TN, 6 MN, and 2 NN were profiled. Among the TN, none possessed class CD4-bs neutralizing antibodies. 622800 possessed PGT128-like response (likely from BG18). Conversely, 3 of the MN possessed CD4bs antibodies. Thus, not unexpectedly, we were not able to identify an enrichment of known bNAb sequences in our cohort.

Conversely, it is important to note that BG18 and 3BNC117 were isolated from controllers, supporting our findings that extended SHM and affinity maturation occurs in these unusual individuals, and interestingly leads to the evolution of unique clonal lineages that give rise to potent broadly neutralizing antibodies – even in the absence of substantial antigenemia.Finally, we agree 100% with the reviewer that looking at CDR3 length alone is not an adequate predictor of neutralization capacity. We have made this explicitly clear in the text. However, for simple descriptive reasons we still compared CDR length between TN and NN for completeness. However, instead of comparing length means, we deployed a Kolmogorov-Smirnov test to describe length distributions in the groups. Indeed, using this approach, and as suspected by the reviewers, the 2 groups had significantly different CDR lengths.

2. It is unclear whether the broad neutralizing activities in the samples examined here targeted a single or multiple envelope epitopes. The authors should explicitly discuss this point in the Discussion section.

We agree that this is an important point and now specifically highlighted it in the result section and the discussion.

3. Several parts of the discussion hard to read and confusing. This reflects the somewhat confusing and contradictory findings of high levels of SHM, but contracting evolution that is shown in Figure 3. The discussion of this could do with some tightening. Parts are speculative and wordy, e.g., the first paragraph of page 8 could be shortened.

We apologize if the discussion has led to confusion and have now modified and shortened the text. We hope that the findings are now better explained, specifically emphasizing the different phases during the selection process with slowing down of the continuous accumulation of additional random amino acid changes once a certain level of affinity has been reached, that explain some of the observations in Figure 3.

4. It is unclear whether a cocktail of the five different Envs listed in the Methods section was used to sort B cells, and if the same Envs were used in every case or not.

Yes, a cocktail of the five different Envs listed were used in every case and we have clarified this in the methods section.

5. The manuscript reports that the difference in CDRH3 length distribution (Figure 1F) is not statistically significant (they do not report the exact p value produced by the Wilcoxon rank sum test). However, the difference appears substantial in Figure 1F. One wonders whether a different statistical test or looking at subpopulations might give a statistically-significant result. Also, in Figure 1F, can the authors comment on the very high frequency of sequences in NN with CDRH3 lengths of 9 and those of 16 on the TN? Are these sequence artefacts, perhaps of oversampled clones in certain donors?

We now use the Kolmogorov-Smirnov test to compare CDRH3 length distributions in NN and TN rather than comparing the median CDRH3 lengths. Indeed, the CDRH3 length distribution in TN sequences was significantly different to CDRH3 length distribution in NN sequences (*p* < 2.2e-16 Kolmogorov-Smirnov test; Figure 1C). Furthermore, significant differences in length were observed for CDRH2, CDRL1, CDRL3 (*p* < 2.2e-16 Kolmogorov-Smirnov test). There were no significant differences for CDRH1 and CDRL2. As correctly assumed by the reviewer, 355 of 488 sequences in NN with CDR3 lengths of 9 belong to the VH4-34/J4 – KV2-24.2D/KJ1 clone from donor 211774, while 978 of 1355 sequences in NN with CDR3 lengths of 16 belong to the VH1-24/J4 – LV1-51/LJ1 clone from donor 330183. As now we present the CDR3 length distributions separately for each individual and test the difference between the distributions using Kolmogorov-Smirnov test rather than comparing only medians with Wilcoxon test, we believe that our results no longer represent artifacts of oversampled clones.

6. The authors state that sequence data will be deposited in GEO. The processed sequences (used to make Figure 2, etc.) should be made available, not just the raw NGS sequencing.

We agree with the reviewer and have instead attached “Supplementary file 2” with all annotated sequences.

7. There is some inconsistency with the V gene names – "VH1-2" vs. "HV4-59".

All names are corrected to be consistent in the “HV” format.

8. The proposal in the Introduction that neutralizing breadth may be achievable in the absence of high levels of viremia is not entirely new, as broadly neutralizing antibodies have been isolated from elite controllers. For instance, Kwong and colleagues reported on the presence of broadly neutralizing antibody responses in a cohort of long-term non-progressors (Gonzalez et al. PLOS ONE 2018) and the Bjorkman group reported on the isolation of a broadly neutralizing monoclonal antibody from a 'controller' (Schoofs et al. Immunity 2019).

We have modified the language in the introduction and have added the 2 references.

9. Page 4 – The authors discuss the varying Ig subclass distributions and suggest for example that the higher percentage of IgG3 in NN versus TN might suggest a higher proportion of more recently induced MBCs in the NNs. These differences are not very convincing, if anything they are driven by a single data point, and we suggest removing related discussion.

The respective discussion has been removed.

10. Figure 1F – Can the authors comment on the very high frequency of sequences in NN with CDRH3 lengths of 9 and those of 16 on the TN. Are these sequence artefacts, perhaps of oversampled clones in certain donors?

We have modified the analysis for CDR length comparisons and now use the Kolmogorov-Smirnov test to compare length distributions in NN and TN rather than comparing median lengths (Figure 1C). As correctly suggested by the reviewer, 355 of 488 sequences in NN with CDR3 lengths of 9 belong to the VH4-34/J4 – KV2-24.2D/KJ1 clone from donor 211774, while 978 of 1355 sequences in NN with CDR3 lengths of 16 belong to the VH1-24/J4 – LV1-51/LJ1 clone from donor 330183. The new analysis present CDR3 length distributions separately for each individual and test the difference between the distributions using Kolmogorov-Smirnov test rather than comparing only medians with Wilcoxon test, we believe that our results no longer represent artifacts of oversampled clones.

11. Page 5 – could the authors translate frequency of nucleotide mutations into an approximate % SHM for easy comparison to other studies particularly of bNAbs. How close are these mutation counts of >60 to the levels of SHM we associate with neutralization breadth?

Translated mutation frequencies are added along with the mutation counts. Mutation frequencies were calculated by comparing the entire input sequence to the germline sequence. Figure 2A,D,G,H,I were modified to depict the %SHM. Previously defined >60 mutation count cutoff corresponds to %15 SHM and >40 mutation count to %10 SHM. Figure 3A is replotted using %SHM. This is consistent with the spread of SHM rates in known bNAbs that can range from 10 to 30% (ie VRC01 class antibodies).

12. Figure 2 – In both panels A (IGHV) and C (IGLV), while the NNs largely cluster towards the left (with low numbers of mutations), there are a cluster of NN sequences with higher numbers of mutations (~80 for IGHV and 50 for IGLV). Are these sequences from specific donors? What, if anything, distinguishes them from the other sequences?

Reviewers suggestion of translating mutation counts to frequencies helped us resolve the cluster of NN sequences with higher numbers of mutations (~80 for IGHV and 50 for IGLV). We now present Figure 2A and 2C panels as %SHM frequency distributions for each individual separately. As NN sequences with higher mutation counts were longer sequences, translated mutation frequencies do not show these clusters of NN sequences anymore.

13. Page 6 – the last statement that the first paragraph seems somehow circular – "Assuming that the somatic IGHV mutation rates in B cells correlate with the total exposure time in the GC environment, and in light of the fact that the TN and NN did not differ in age, time-off-ART and viral loads, our data suggests that in the TNs, B-cell clones were selected for extensive SHM." How would B cell clones be selected FOR extensive SHM?

We agree with the reviewer that this is misleading and have modified the text to state that “…TNs were more permissive for extensive SHM of B-cell clones”.

14. Page 6 – Clones may show variation in terms of CDRH3 length – did the authors explore the effect of softening these criteria e.g. by allowing +/- 1 amino acid on clone size?

We appreciate the suggestion, and we repeated the clonal inference allowing +/- 1 CDRH3 amino acid length. We would expect that softening these criteria could decrease the number of unique clones identified. However, we did not see a change in clone sizes. We evaluated various options available within the Immcantation framework (the computational pipeline we chose for this study), which explores clustering thresholds and distance measures for CDRH3 sequences. We finally selected the default Hamming distance threshold as explained in the methods.

15. Figure 3 – legend refers to a mean number of sequences per clone of 7.7 that differs from the text.

These numbers have been corrected to be consistent with the recent clonal analysis. Mean number of sequences per clone was changed to 7.4.

16. Page 7, third line, first para of discussion – why "in contrast"?"Moreover, the mutation frequency in our data correlated directly with the neutralization breadth suggesting a skewing towards poly- or cross-reactive antibodies that are able to bind mutated forms of the original epitope" – do high mutation frequencies necessarily translate to polyreactivity?

We have modified the discussion to focus more on the paired IGH/IGL analysis and now highlight the differences observed. This has changed the first paragraph of the discussion substantially. With regards to the link between high mutation rates and poly-reactivity we agree with the reviewer that high mutation frequencies rather increase affinity than polyreactivity and have changed the text accordingly.

17. In the Results section, at the end of clonal selection section (page 7), the discussion of "maximum affinity maturation" does not seem to reckon with the continuous change in the Env antigen, which generally manages to escape antibody pressure.

We agree with the reviewer that this is reads confusing and have modified the text to reflect that the process of affinity maturation has slowed down (rather than reached a maximum), and that the continuous accumulation of additional random amino acid changes would rather be detrimental and decrease affinity.

18. Page 8 – correct spelling of "alternatively"

The spelling is corrected.

19. Page 2 and page 20 – Supplemental Table 1. Why have the authors used a background of 3x MuLV? This is different from most studies of neutralization breadth. Why do some donors have activity against MuLV? What contributes to this non-specific background? Also, the legend contains a typo – please correct.

The non-specific background activity that was observed when using negative control virus MuLV was most likely due to anticoagulants that are present in the patient plasma samples. All anticoagulants can cause some level of cell stress/toxicity of TZM.bl cells at low plasma dilutions. While serum is always preferred for measuring neutralizing antibodies in the TZM.bl assay to avoid this issue, plasma samples are also acceptable when proper controls are in place to account for any non-specific activity, which may also vary between individual samples. The use of a 3-fold cut-off for determining positive responses is actually a standard practice in the field. The validation criteria for the TZM.bl assay outline that titers within a 3-fold range can be considered equivalent (Sarzotti-Kelsoe, M. et al., 2014, J. Immunol. Methods, 409: 131-146). We have referenced this paper in the methods description. Thus, it is critical that a stringent cut-off (i.e. > 3-fold difference in ID50 titers) is utilized for establishing positivity criteria when comparing neutralization responses against HIV-1 pseudovirus strains versus non-specific background observed against non-HIV negative control viruses.